# Direct measurement of the mechanical work during translocation by the ribosome

Tingting Liu[1,2†‡], Ariel Kaplan[1,2,3,4*†], Lisa Alexander[5], Shannon Yan[5], Jin-Der Wen[5§], Laura Lancaster[6,7], Charles E Wickersham[1,2], Kurt Fredrick[6,7¶], Harry Noller[6,7], Ignacio Tinoco Jr[5], Carlos J Bustamante[1,2,5,8,9*]

[1]Jason L Choy Laboratory of Single Molecule Biophysics, University of California, Berkeley, Berkeley, United States; [2]Department of Physics, University of California, Berkeley, Berkeley, United States; [3]Faculty of Biology, Technion-Israel Institute of Technology, Haifa, Israel; [4]Lorry I Lokey Interdisciplinary Center, Technion-Israel Institute of Technology, Haifa, Israel; [5]Department of Chemistry, University of California, Berkeley, Berkeley, United States; [6]Department of Molecular, Cell, and Developmental Biology, University of California, Santa Cruz, Santa Cruz, United States; [7]Center for Molecular Biology of RNA, University of California, Santa Cruz, Santa Cruz, United States; [8]California Institute for Quantitative Biosciences, University of California, Berkeley, Berkeley, United States; [9]Department of Molecular and Cell Biology, Howard Hughes Medical Institute, University of California, Berkeley, Berkeley, United States

*For correspondence:
akaplanz@technion.ac.il (AK);
carlos@alice.berkeley.edu (CJB)

†These authors contributed equally to this work

Present address: ‡Cell Biology and Biophysics Unit, Porter Neurosciences Research Center, National Institute of Neurological Disorders and Stroke, Bethesda, United States; §Institute of Molecular and Cellular Biology, National Taiwan University, Taipei, Taiwan; ¶Department of Microbiology, The Ohio State University, Columbus, United States

Competing interests: The authors declare that no competing interests exist.

**Abstract** A detailed understanding of tRNA/mRNA translocation requires measurement of the forces generated by the ribosome during this movement. Such measurements have so far remained elusive and, thus, little is known about the relation between force and translocation and how this reflects on its mechanism and regulation. Here, we address these questions using optical tweezers to follow translation by individual ribosomes along single mRNA molecules, against an applied force. We find that translocation rates depend exponentially on the force, with a characteristic distance close to the one-codon step, ruling out the existence of sub-steps and showing that the ribosome likely functions as a Brownian ratchet. We show that the ribosome generates ~13 pN of force, barely sufficient to unwind the most stable structures in mRNAs, thus providing a basis for their regulatory role. Our assay opens the way to characterizing the ribosome's full mechano–chemical cycle.

## Introduction

Ribosomes possess three binding sites for tRNA: the aminoacyl (A), peptidyl (P), and exit (E) sites, each of which is shared between the 30S and 50S ribosomal subunits. Following codon recognition and peptide bond formation, the ribosome has a deacylated tRNA in the P site and a peptidyl-tRNA in the A site. In order to start a new elongation cycle, the A site must be emptied to allow binding of the next aminoacyl-tRNA. To this end, the tRNAs and mRNA must move relative to the ribosome. This movement occurs in two steps (*Moazed and Noller, 1989b*): first, the 3′ ends of the tRNAs in the A and P sites move, with respect to the 50S subunit, into hybrid A/P and P/E states, respectively. In vitro, formation of these states can occur spontaneously, reversibly, and independently of elongation factor G (EF-G) (*Moazed and Noller, 1989b*; *Sharma et al., 2004*; *Cornish et al., 2008*; *Munro et al., 2010*) and is coupled to rotation of the 30S body (*Moazed and Noller, 1989b*; *Frank and Agrawal, 2000*; *Agirrezabala et al.,*

**eLife digest** Producing a protein first requires its gene to be transcribed into a long molecule called a messenger RNA (mRNA). A complex molecular machine called the ribosome then translates the mRNA code by reading it three letters at a time. Each triplet of letters—known as a codon—tells the ribosome which amino acid to add next into the protein. After adding an amino acid, the ribosome moves along the mRNA molecule to read the next codon and add another amino acid into the protein chain.

While researchers understand how protein chains are formed, how the ribosome shifts along the mRNA strand—a process called translocation—is still unclear. It is known that this process involves many force-generating movements and changes to the shape of the ribosome. However, it is only recently that researchers have been able to measure these forces.

Using optical tweezers—an instrument that uses a highly focused laser beam to hold and manipulate microscopic objects—Liu, Kaplan et al. followed individual ribosomes as they translated an mRNA and measured the effect that applying an opposing force has on the rate of translation. The results shed new light on the mechanism of translocation. First, Liu, Kaplan et al. found that ribosomes jump directly from one triplet to the next in the mRNA sequence, rather than moving there in a series of smaller steps. Next, the results indicate that translocation occurs spontaneously, driven by thermal energy, while chemical reactions prevent the reverse movement, in a mechanism known as a 'Brownian Ratchet'.

Measurements of the maximum force generated by the ribosome also give insights into how translation is regulated. Strands of mRNA can fold into certain structures that slow down translation, because the mRNA must first be unfolded before the ribosome can translate it. Liu, Kaplan et al. found that the maximum force generated by a ribosome is only just enough to unwind these mRNA structures, making the translation rate highly sensitive to the existence of such structures, and the structures themselves of high importance for regulating transcription.

Given its importance as the ultimate decoder of the genetic information, understanding the ribosome's function and regulation has broad implications. The work of Liu, Kaplan et al. opens the way for a full characterization of the role of mechanical forces in the translation process.

*2008*; *Julian et al., 2008*; *Dunkle et al., 2011*). In the second step, which is irreversible and EF-G-dependent (*Moazed and Noller, 1989b*; *Savelsbergh et al., 2003*), the mRNA is translocated by one codon, along with movement of the associated anticodon ends of the tRNAs to the classical P and E sites, coupled to an orthogonal rotation of the 30S subunit head domain (*Ratje et al., 2010*; *Dunkle et al., 2011*; *Ermolenko and Noller, 2011*; *Guo and Noller, 2012*; *Zhou et al., 2013*). The translocation process also involves other large-scale conformational changes in the ribosome, including reverse rotational movements of the 30S subunit body and head (*Ermolenko and Noller, 2011*; *Guo and Noller, 2012*), and movement of the large subunit L1 stalk into the intersubunit space (*Fei et al., 2008*; *Cornish et al., 2009*). Translocation is therefore a highly coordinated and complex process composed of inter- and intra-molecular, force-generating mechanical movements.

During translation, the ribosome must also overcome significant mechanical barriers posed by structured portions of the mRNA. These structures are exploited by the cell to create diverse strategies for translation regulation; for example, pseudoknots and hairpins are used to induce programmed frameshifting (*Tsuchihashi, 1991*; *Namy et al., 2006*), whereas synonymous mutations that can alter the local structure of RNA and codon usage are employed to control protein expression levels (*Duan et al., 2003*; *Nackley et al., 2006*). Hence, force is not only a product of the chemical reactions during translation, but also an important player in the regulation of this process.

## Results and discussion

We have designed an experiment to monitor the movement of individual ribosomes against an opposing force during translation (*Figure 1A*). A gene fusion encoding ribosomal protein S16 linked via its C-terminus to the biotinylation domain of the biotin carboxyl carrier protein (BCCP) was introduced (*Link et al., 1997*) into the chromosome of *Escherichia coli*, and biotinylated ribosomes were then purified. In the presence of initiation factors, initiator tRNA and GTP, a biotinylated ribosome is assembled at

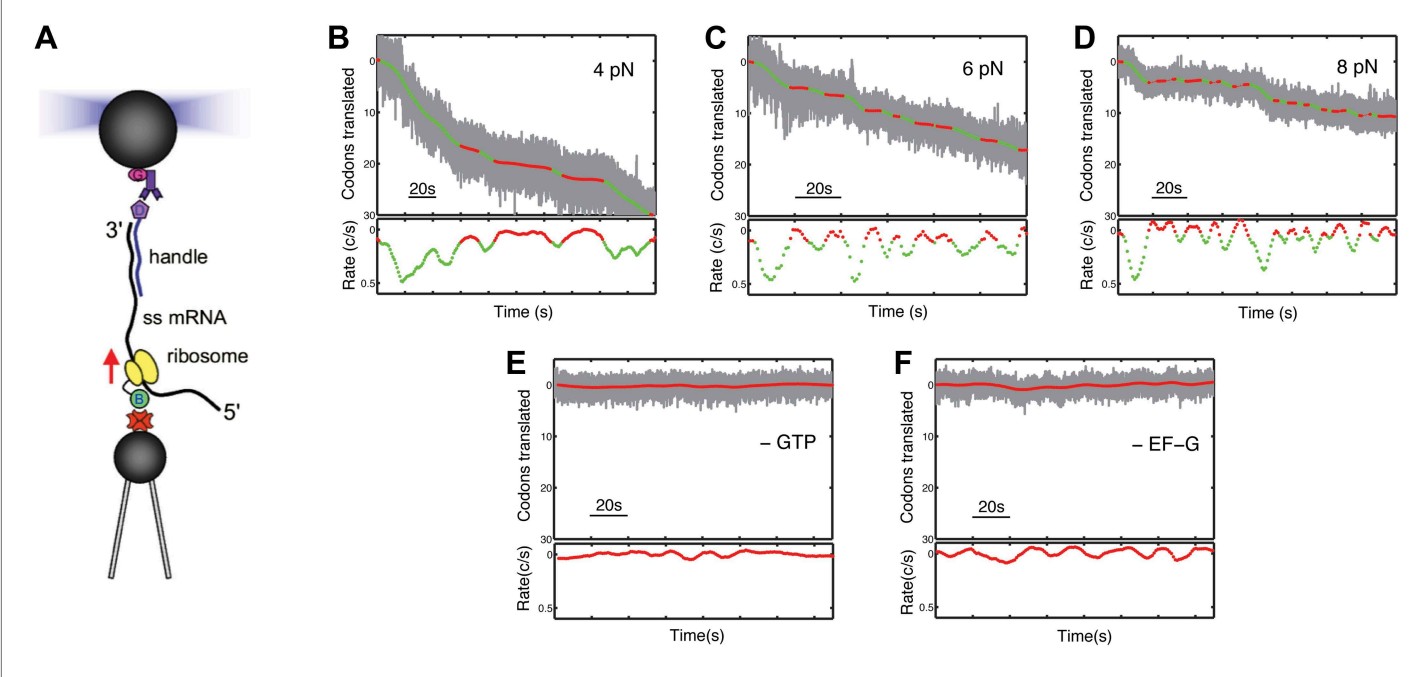

**Figure 1**. Following translation by a single ribosome on a single mRNA. (**A**) Geometry for single-molecule translation experiments. A biotinylated ribosome is loaded onto a single-stranded mRNA and attached to a streptavidin-coated polystyrene bead fixed to a micropipette. The 3′ of the message is anchored to a second bead through a 1460 bp DNA/RNA hybrid handle. Calibrated forces can be applied to the ribosome by manipulating the second bead with an optical trap, while the translation progress of the ribosome is determined by the change in extension of the tether. (**B–D**) Typical translation events recorded under 4, 6 and 8 pN of constant tension. The upper panels show the codons translated as a function of time, and indicate that translation proceeds not in a continuous manner, but in a series of translational bursts separated by long pauses. The gray line shows the raw (1 kHz) data, while green (translocation) and red (pause) are filtered down to 1 Hz. The lower panels show the instantaneous velocities calculated from the traces above. (**E** and **F**) Control experiments, under 8 pN of tension, showing that in the absence of GTP or EF-G, no translation signals were detected.

The following figure supplement is available for figure 1:

**Figure supplement 1**. A partial translation trace showing an unusually low noise level, and a sequence of presumptive single-codon translocation steps.

the AUG start site of an mRNA, whose 3′ end had been previously annealed to a complementary DNA handle harboring a 5′ digoxigenin. The complex is then tethered between a pair of 2.1 µm diameter polystyrene beads: a streptavidin-coated bead, which binds to the ribosome and is held by suction on the end of a micropipette, and an anti-digoxigenin antibody-coated bead, which binds to the DNA handle and is held in an optical trap. Next, a mixture containing elongation factors, aminoacyl-tRNAs and GTP is introduced into the experimental chamber. The tension in the tether, stabilized by an automated feedback routine, produces a constant opposing force as translocation proceeds. Translation is followed in real time as a decrease in the tether length between the beads (**Figure 1B–C**, **Figure 1—figure supplement 1**). No translation signals were detected in the absence of GTP and EF-G. (**Figure 1E,F**).

Ribosomes do not translate the message in a continuous manner, but in bursts of translation separated by long pauses (**Figure 1B**) that do not appear to be correlated with template position. As in all single-molecule experiments, there is a distribution of the noise level in the different single ribosome trajectories. As a result, some of the trajectories observed during translation bursts exhibit a particularly low noise level, and clearly show that the ribosome moves in periods of stationary dwells that are followed by translocation events corresponding to single codon steps (three nucleotides) along the mRNA (**Figure 1**, **Figure 1—figure supplement 1**). However, in most trajectories the noise prevents us from unambiguously identifying the individual steps, and we therefore base our analysis on the average properties of the translation bursts. We separate these pauses from active translation bursts using a velocity threshold (see 'Materials and methods'), and calculate the 'pause-free' velocity of the ribosome (**Figure 1C**).

We find that as the opposing force, $F$, is increased, the mean pause-free velocity, $v$, decreases monotonically (**Figure 2**), indicating that mRNA translocation is rate limiting under the conditions of the experiment. Note that the applied force acts between the 3' end of the mRNA and the small 30S subunit; there is no directly applied force between the mRNA and the large 50S subunit or between the 30S and the 50S subunits. In addition, the position of the attachment point (protein S16, in the 'back' of the 30S subunit) was chosen because it is remote of any known functional site and is not known to exhibit conformational dynamics during translocation. As a result, force only affects directly the mechanical step in which the anticodon loops of the tRNAs move from the A and P sites to the canonical P and E sites of the 30S subunit, respectively, together with the concomitant movement of the mRNA with respect to the ribosome. Hence, the position of the ribosome relative to the mRNA provides a convenient reaction coordinate to follow the translocation reaction during translational elongation. We can, thus, fit our data to an Arrhenius expression

$$v(F) = v_0 \exp\left(-\frac{F \cdot \tilde{x}}{k_B T}\right),\tag{1}$$

where $\tilde{x}$ is the typical distance over which the force acts, $v_0$ is the zero-force translocation velocity, $k_B$ is Boltzmann's constant and $T = 296$ K is the absolute temperature. The fit yields a zero-force velocity $v_0 = 2.9$ codons/s (1.8, 4.0) and a distance $\tilde{x} = 1.4$ nm (0.9, 1.8). The numbers in parenthesis indicate 95% confidence bounds.

Notably, these results have implications for the intrinsic step size during translocation: crystal structures (**Jenner et al., 2010**) show that the distance between A- and P-site mRNA codons equals 1.48 nm. Clearly, at the end of translocation the mRNA has moved by this distance from its pre-translocation position; however, this value can either reflect a single, one-codon mechanical movement (a single barrier crossing in the free energy surface landscape) or successive smaller substeps (multiple barrier crossings) that sum to one codon, for example, three one-nucleotide substeps. Directly observing these potential substeps requires a temporal and spatial resolution that is not possible in our present experiment because the flexible nature of the mRNA and the relatively low forces involved give rise to high levels of thermally-induced fluctuations. However, because the distance $\tilde{x}$ determined here is similar to the measured one-codon translocation step, we can rule out one- and two-nucleotide translocation substeps and conclude that codon translocation is performed by the ribosome in a single step.

In addition, our measurements shed light on the mechanism of translocation. Mechano-enzymes in general act by coupling a mechanical task (translocation, force generation, work) to a downhill chemical reaction (i.e., a reaction that lowers the total free energy of the system) (**Bustamante et al., 2004**). Clearly, given the diversity of conformational changes and chemical events associated with translocation by the ribosome, a complete description of the process should involve diffusion on a free-energy hypersurface with high dimensionality. However, given that our attachment geometry ensures that we affect and probe a single and well-defined mechanical coordinate, we can reduce this description to a simplified two-dimensional picture. In this two-dimensional free energy landscape one axis represents the mechanical coordinate that describes the movement of the mRNA relative to the 30S subunit, and the other axis the chemical coordinate that describes all binding, hydrolysis and dissociation processes (**Figure 3**) and, for the sake of simplicity, also conformational changes that have a reaction coordinate orthogonal to the reaction coordinate probed in our experiments. The most likely path for the reaction occurs along a minimum energy channel on this surface and the different events involved in translocation can now be described as diffusive

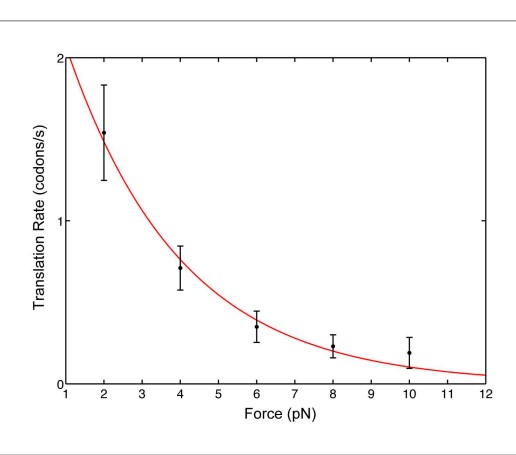

**Figure 2**. Pause-free translational velocity as a function of opposing force. Data points are the mean velocities for all measured traces at each force (N = 54). Error bars represent the standard error of the mean. The solid line is an exponential fit of the form $v(F) = v_0 \exp\left(-\frac{F \cdot \tilde{x}}{k_B T}\right)$.

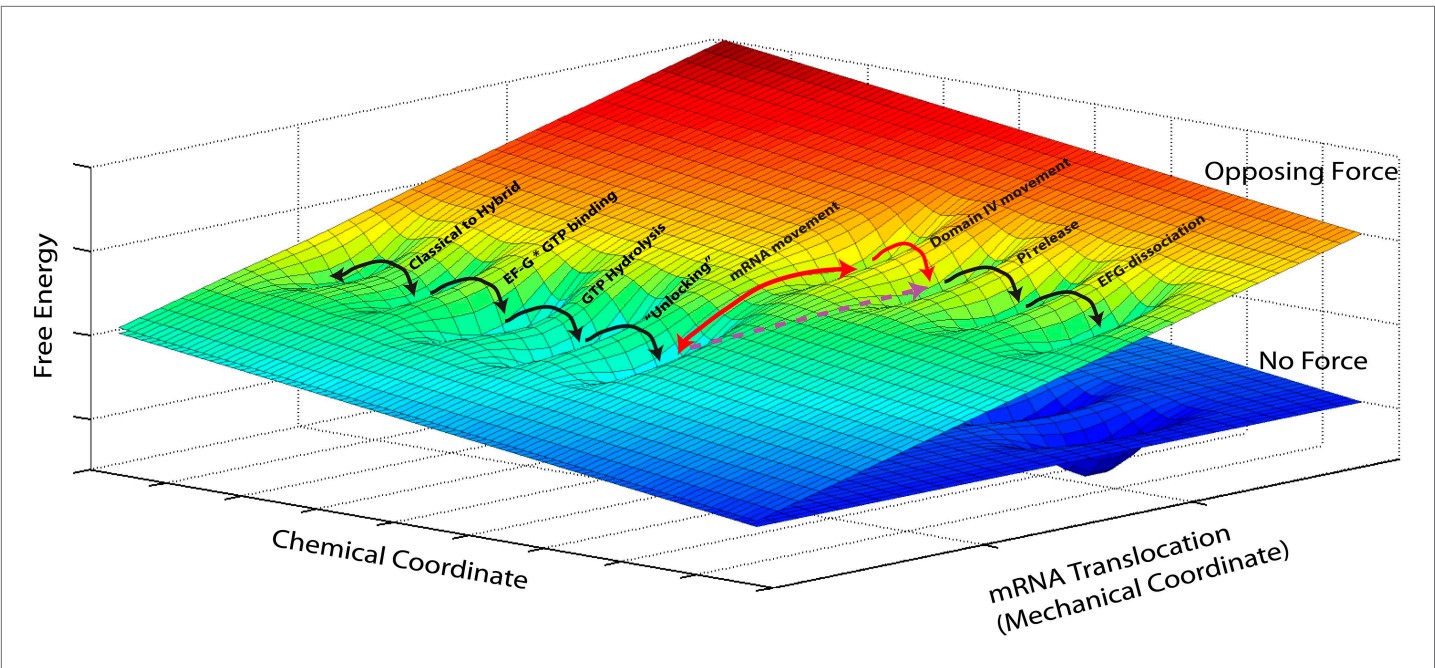

**Figure 3**. Reduced energy landscape for mRNA translocation. The mechanical coordinate describes the movement of the mRNA relative to the 30S subunit, while the chemical coordinate describes all binding, hydrolysis and dissociation processes, in addition to conformational changes with a reaction coordinate orthogonal to the translocation coordinate probed in our experiments. Translocation proceeds by diffusive transitions between minima of this reduced energy surface. A Power Stroke mechanism involves a diagonal transition, with simultaneous progress in the chemical and mechanical axis (dashed, purple line). Alternatively, a Brownian Ratchet (full, red lines) is composed of two orthogonal transitions: a fast equilibrium between pre- and post-translocated states along the mechanical coordinated, followed by a 'rectifying' chemical transition.

transitions between minima of this (reduced) energy surface. Thus, for example, the classical-to-hybrid transitions and the associated ribosomal intersubunit rotations are assigned as movements along the chemical coordinate over a rather shallow activation energy that accounts for their reversible nature (***Munro et al., 2007***; ***Cornish et al., 2008***; ***Fei et al., 2008***). The three-dimensional energy surface depicted in ***Figure 3*** naturally explains how transition rates are affected when a mechanical force is applied. The effect is equivalent to tilting the potential energy surface by rotating the diagram around its chemical axis (***Bustamante et al., 2004***), hence affecting the rate and equilibrium constants of reactions along the mechanical coordinate, for example making translocation more (force applied in the aiding or 'pushing' direction) or less (force applied in the opposing or 'pulling' direction) favorable.

Translocation of mRNA and its two associated tRNA anticodon stem-loops from the A and P sites to the P and E sites of the 30S subunit (a movement along the 'mechanical' axis) must then be coupled to a downhill progress along the 'chemical' axis. Fundamentally, there are two ways in which this coupling can occur: one possibility is that the energy released by the 'chemical' transition is directly harnessed to produce the change. In this case, usually called a 'Power Stroke (PS)' mechanism, the system moves diagonally in the energy landscape. Alternatively, it is possible that the system moves back-and-forth spontaneously, driven by thermal energy, along the mechanical coordinate, until a chemical transition, that occurs when the system is in the post-translocated state, prevents the back-translocation and 'rectifies' this random motion into directed motion. This second mechanism, in which the system moves on the energy landscape in two orthogonal steps, is called a 'Brownian Ratchet (BR)'. Importantly, although both these different molecular mechanisms will result in a velocity which depends exponentially on the force, as in the Arrhenius equation above, they will differ in the identity of the distance $\tilde{x}$ (***Wang et al., 1998***): in the PS case, as the post-translocation state is achieved in a single (diagonal) transition, the force-dependence of the reaction rate will be given by an Arrhenius expression in which $\tilde{x} = x^{\dagger\dagger}$, that is, the distance along the mechanical coordinate to the transition state during translocation (***Bustamante et al., 2004***). Alternatively, in the BR case, the post-translocation state is achieved

via two transitions, and the rate will be given by the product of the (force-dependent) probability of spontaneously populating the post-translocation state and the (force-independent) rate of the chemical reaction. As a result, the force dependence of the velocity is dictated by the force-dependent equilibrium constant between the pre- and post-translocated states, and hence described by an Arrhenius expression in which $\tilde{x} = x_{step}$, that is, equal to the distance between these states, or the step size of the motor (*Bustamante et al., 2004*). While these are two idealized cases and a real system is likely to combine features of these two models, the distance determined here (1.4 nm) is very close to the full 1.48 nm step size and indicates that, while we cannot fully rule out a PS mechanism, the ribosome likely functions as a BR during translocation. Notably, recent crystallographic studies of the ribosome bound to EF-G in a translocation intermediate (*Chen et al., 2013*; *Pulk and Cate, 2013*; *Tourigny et al., 2013*; *Zhou et al., 2013*), and a previous structure of the ribosome bound to EF-G in the post-translocation state (*Gao et al., 2009*) can lend support for this result and help clarify the identity of the reaction that rectifies translocation in this ratchet mechanism: the structures suggest that domain IV of EF-G could prevent back-translocation of the P-site tRNA by occupying the A-site, and that intercalation of two highly conserved bases of 16S rRNA into mRNA could prevent its back-translocation. These conformational changes could thus act as 'pawls' in the BR mechanism. Interestingly, the fact that both tRNA and mRNA movements would be locked could contribute to prevent frame shifting.

The force at which the velocity approaches zero (the stall force) represents the maximum force that can be intrinsically generated by the motor in a cycle. Our results indicate that the ribosome can generate forces as high as 13 ± 2 pN. Remarkably, the ribosome stall force is very close to that required to unwind the strongest secondary structure motifs typically present in mRNA (*Tinoco et al., 2004*). The comparable magnitude between the mechanical strengths of RNAs and the stall force of the ribosome indicates that RNA secondary structures can have a strong effect on the rate of translation (and hence on phenomena such as frameshifting and cotranslational folding of the protein) and explains how these structures, while not being insurmountable barriers, can fulfill a regulatory role in the cell.

The work generated by the ribosome near stalling, that is, the product of the stall force and the step size, 21.2 pN · nm = 5.2 kBT or about 3.1 Kcal/mol, represents the maximal mechanical work generated by the motor during the translocation step. What is the energetic source for this mechanical work? The free-energy difference between peptide bond and ester bond hydrolysis is approximately −3.7 ± 1.2 kcal/mol, equivalent to 6.3 ± 2 kBT per bond exchange in our experimental conditions ('Materials and methods'). Hence, the maximal mechanical work that can be generated by the ribosome is ~80% of the total energy available from transpeptidation and, in principle, it is possible to power translocation from this energy without the need to invoke an energetic contribution from the hydrolysis of GTP. In fact, studies have shown that the ribosome can translocate in the absence of EF-G (*Gavrilova et al., 1976*) or in the absence of GTP (*Pestka, 1969*; *Rodnina et al., 1997*; *Fredrick and Noller, 2003*). However, spontaneous forward translocation is unfavorable in many contexts (*Shoji et al., 2006*), and efficient and rapid translocation does require EF-G and GTP hydrolysis. Furthermore, 80% thermodynamic efficiency for conversion of chemical energy to mechanical motion is higher than occurs in most molecular motors (*Bustamante et al., 2004*), and, moreover, it is not clear how the energy available from peptide bond formation could be stored and transmitted from the 50S to the 30S subunit. Instead, in view of our results, a mechanism in which EF-G binding and GTP hydrolysis account for the energy of translocation and resetting (including EF-G–GDP dissociation) appears to be more likely.

Not surprisingly, the mechanism of action of the ribosome as a mechano-enzyme during translocation is more complicated than that of a typical molecular motor. One possible scenario that emerges from this and previous studies is that translocation is achieved by two consecutive BRs: during the first step, the ribosomal subunits rotate back and forth relative to each other along an axis perpendicular to the subunits interface. This process, reversible and thermally activated, is accompanied by the repositioning of the 3′-acceptor ends of the tRNAs initially in the classical A and P states into the hybrid A/P and P/E states and movement of the L1 stalk. Binding of EF-G–GTP stabilizes the tRNAs in their hybrid states and the counter-clockwise rotation of the 30S subunit relative to the 50S subunit. Hence, binding of EF-G functions as a rectifying reaction for the first BR. The ribosome acts as a GTPase activator for EF-G, and rapid GTP hydrolysis catalyzes conformational changes in the ribosome (e.g., swiveling of the head, which opens the way for the passage of the P-site tRNA anti-codon stem-loop to the E site) that result in allowing spontaneous and thermally activated transitions between the pre- and post-translocation

state. This second BR is likely to be rectified by the movement of domain IV of EF-G into the A-site and the intercalation of two conserved bases of 16S rRNA into mRNA. Pi release then induces a relaxation of EF-G, destabilizing the contacts between domains III and V and the ribosome and resulting in EF-G dissociation.

Finally, we turn our attention now to the pauses observed during translation. The duration of the translation bursts (and hence the effective pause entry rate, $k_p$) is independent of the force opposing translocation (*Figure 4A*). Moreover, the full distribution of burst durations is well described, at all forces, by a single exponential function with nearly identical parameters (*Figure 4—figure supplement 1*). Likewise, *Figure 4B* shows that the duration of the pauses (and hence the pause exit rate, $k_{-p}$) is independent of force. Since the pause duration is also well described by single exponential functions with the same parameters at all the tested forces (*Figure 4—figure supplement 1*), we conclude that entering into the paused state and exiting from it are both governed by single, force-independent steps. Hence, we can cluster all our measurements into one data set and calculate the entry and exit rates: $k_p = 0.16$ s$^{-1}$ (0.14, 0.18) and $k_{-p} = 0.14$ s$^{-1}$ (0.1, 0.18), where the numbers in parenthesis indicate 95% confidence intervals.

What is the origin of the observed pauses? It is well known that translating ribosomes tend to pause at specific secondary structure motifs (such as hairpins and pseudoknots). However, if that were the case for the observed pauses, their occurrence would be correlated with the position on the template (which we do not observe) and their density would depend strongly on the applied force, as this force will destabilize the secondary structure ahead of the ribosome in the geometry used in these experiments. Thus, the force independence of the entry and exit rates rules out this possibility. Hence, we favor a model in which these pauses represent events that are off-pathway from the main incorporation and translocation cycle. The rate limiting transitions into these off-pathway states are steps along a non-mechanical reaction coordinate, or alternatively a mechanical coordinate that is orthogonal to the translocation movement, and hence not affected by the externally applied force. Interestingly, the pause entry and exit rates are very similar, indicating that in the conditions of our experiments the time spent by the ribosome in the productive translation states and the unproductive paused state are nearly the same.

The geometry of our experiments defines a single reaction coordinate. A more complete description of the translation process will require a multidimensional energy space. Nonetheless, the assay presented here opens the way for additional experiments where, by choosing different attachment points on the ribosome—across the small and large subunits for example—it may be possible to mechanically probe the internal degrees of freedom associated with ribosome translocation. Such experiments should reveal how this motor coordinates its internal dynamics with its translocation and helicase activities.

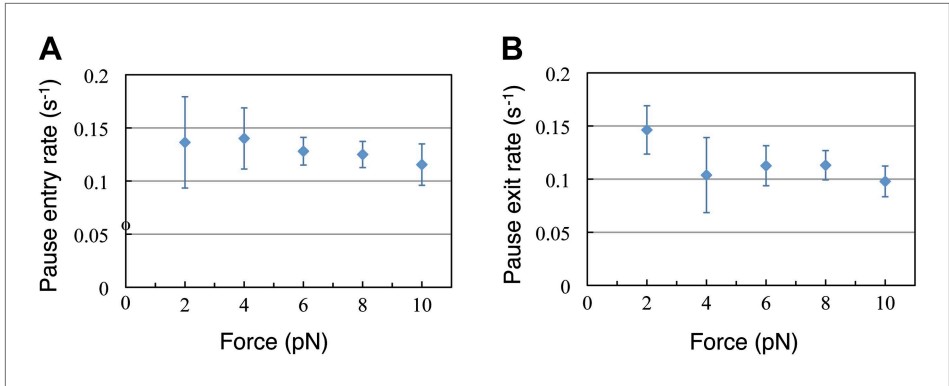

**Figure 4**. Pause entry and exit rates. (**A**) Pause entry rate, calculated as the inverse of the mean duration of the translation bursts in between the pauses. (**B**) Pause exit rate, equal to the inverse of the mean pause duration. Both rates are essentially independent of the applied opposing force.
The following figure supplement is available for figure 4:

**Figure supplement 1**. Distribution of the translation bursts and pauses durations for all the measured opposing forces.

## Materials and methods

### Construction of *E. coli* strains with biotinylated ribosomes

A gene fusion encoding S16 linked via its C-terminus to the biotinylation domain of the biotin carboxyl carrier protein (BCCP) was introduced into the chromosome of *E. coli*, using the allelic replacement method of Church et al. (*Link et al., 1997*). SDS-PAGE analysis of 30S subunits purified from these strains indicated the absence of native S16 and the presence of S16-BCCP in stoichiometric amounts. Addition of excess avidin prior to electrophoresis resulted in complete shift of the S16-BCCP band to that of a high molecular weight complex, indicating highly efficient biotinylation of the fusion protein in vivo. Cells expressing the BCCP-ribosome construct (*E. coli* strain KLF203) have no detectable phenotype.

### Purification of ribosomes

Typically, 1 liter of cells grown to mid-log at 37°C in LB broth, was pelleted, resuspended in buffer A (20 mM Tris-Cl [pH 7.0], 100 mM $NH_4Cl$, 10 mM $MgCl_2$, and 5 mM βME), lysed, layered onto a 35 ml cushion containing 1.1 M sucrose, 20 mM Tris-Cl (pH 7.0), 500 mM $NH_4Cl$, 10 mM MgCl2, and 5 mM βME, and centrifuged in a Beckman Ti45 rotor at 36,000 rpm for 21 hr at 4°C. The ribosome pellet was dissolved in 0.5 ml buffer A, layered onto two 38 ml 10–35% sucrose gradients containing buffer A, and centrifuged in a Beckman SW28 rotor at 19,000 rpm for 16 hr at 4°C. The ribosomes in the 70S peak were collected and centrifuged in a Beckman Ti60 rotor at 38,000 rpm for 20 hr at 4°C. The ribosome pellet was dissolved in 0.2 ml buffer A, aliquoted, quick-frozen in liquid $N_2$, and stored at −80°C.

### mRNA synthesis

A DNA oligomer called TTC17, CAACCATGGTCTCG(TTC)17 GTCTTCCTAGGAAC, was synthesized with 17 repeats of the TTC triplet in the center, a BsaI site on the 5′ end and dual BbsI-AvrII sites on the 3′ end. TTC17 was first converted to a double-stranded duplex. Half of TTC17 was cut by BbsI and half by BsaI to remove the sequences after and before the TTC repeats, respectively. Ligation of both restriction fragments (with complementary cohesive ends) resulted in TTC32, which contains 32 TTC repeats with the same flanking sequences as in TTC17: CAACCATGGTCTCG(TTC)$_{32}$GTCTTCCTAGG AAC. The procedure was repeated to generate a sequence (TTC62) with 62 TTC repeats. TTC62 was finally cut with BsaI and AvrII before being inserted into the vector pRC4a, a derivative of pRC4 (*Wen et al., 2008*). pRC4a was then cut with NcoI and AvrII and ligated to an adaptor (CATGCGCTAGCTTA CCATGGGTCTCG) to convert the cohesive end of NcoI to BsaI and thus to allow ligation to TTC62. The plasmid (with 62 TTC repeats) was cut at BspHI and transcribed (Megascript T7 kit, Ambion, Austin, TX) into RNA with a length of 1827 nt. A region (1453 nt) on the 3′ side of the RNA was annealed to a complementary DNA strand (as a 'handle') containing two digoxigenin tags at the end.

### Formation of initiation ribosome-mRNA complexes

To make initiation complex (ICs) the following components were mixed in buffer TL (40 mM HEPES-KOH [pH = 7.5], 60 mM $NH_4CL$, 10 mM Mg [OAc]$_2$, 1 mM DTT, 3.6 mM β-ME) and incubated at 37°C for 15 min: GTP (1.0 mM), mRNA (0.2 μM), initiation factors (IF1, 4.0 μM, IF2, 3.7 μM, IF3, 3.9 μM), fMet-tRNA$^{fMet}$ (3.9 μM) and biotinylated ribosomes (1.0 μM). Finally, 1 μl aliquots of the mixture were prepared, quick-frozen in liquid $N_2$, and stored at −80°C.

### Preparation of translation mixtures

Total tRNA mixtures (Sigma) were aminoacylated using S-100 enzymes (*Moazed and Noller, 1989a*) and extracted with phenol/chloroform. To make a large-scale preparation of EF-Tu·aa-tRNA$^{aa}$·GTP ternary complex, the following components were mixed in a total of 1 ml buffer TL-DTT (buffer TL without DTT): 1 mM GTP, 5 mM PEP, 24 μM EF-Tu, and 0.04 mg/ml pyruvate kinase. The mixture was incubated at 37°C for 15 min. Then, 20 μl (1 U/μl) total aa-tRNAs were added, incubated at 37°C for 5 min, and on ice for 10 min. Since free tRNAs, which are not productive in translation, tend to increase the noise in the translation traces (probably by binding to the single-stranded mRNA), we developed a procedure to purify the ternary complexes using the 6xHis-tag present in EF-Tu. Briefly: The above reaction was bound to a Ni-NTA resin (30 min incubation at 4°C), and the resin washed three times with TL, with the addition of 20 mM imidazole. Ternary complexes were eluted with 600 μl elution buffer (TL + 250 mM imidazole) and dialyzed into Buffer TL for a total of 4 hr. Next, 50 μl purified ternary complexes (containing 0.2 U total aa-tRNA) were diluted with 390 μl (total 440 μl) buffer TL containing

1 mM GTP, 1 mM ATP, 40 U RNAguard (GE Healthcare, Piscataway, NJ) and 1 μM EF-G (final concentrations). Finally, the mixture was filtered with a 0.22 μm low protein- binding MILLEX-GV Durapore membrane, (Millipore, Billerica, MA) and kept on ice.

## Optical tweezers

1 μl of ICs were first diluted with 50 μl TL, and then 1–10 μl mixed with 2.1 μm diameter polystyrene beads coated with anti-digoxigenin antibodies, and incubated on ice for 10 min. During this step, initiation complex attach to the beads through recognition between the complementary DNA handle (harboring a 5′ digoxigenin) that has been annealed to the mRNA and the anti-digoxigenin antibody coated on the beads. The initiation complex beads were then flowed into the chamber of the optical tweezers. After trapping one initiation complex bead in the optical trap, a streptavidin-coated bead, which binds to the ribosome and is held by suction on the end of a micropipette was moved to approach to the initiation complex bead. Binding of the biotinylated ribosome on the initiation complex to the streptavidin bead, results in an initiation complex tethered between two beads. Next, the translation mixture is flown into the chamber, and translation is followed in real time as a decrease in the tether length between the beads. No translation signals were detected before the introduction of the mixture.

The experiments were conducted using force-measuring dual-beam optical tweezers, similar in concept to the instrument described by *Smith et al. (2003)*, but with an improved design that allows for better spatial and temporal resolution. Briefly, the two counter-propagating, orthogonally polarized laser beams that form the trap are coupled into two single-mode optical fibers and focused by two high numerical aperture objectives at a common position. The location of the optical trap can be controlled by tilting the tips of the optical fibers using two piezoelectric crystals, controlled by a feedback loop that maintains the respective focal points at a common position. The position of the trap is measured by a pair of position sensitive detectors (PSDs) that measure the tilting of the beams before entering the objectives, and the force is assessed from a second pair of PSDs to which the light distribution at the back focal planes of the objectives is imaged. All signals are sampled at 1 kHz.

## Data analysis

The 1 kHz raw data of tether extension at constant force was first averaged using a moving Savitzky-Golay filter with a span of 4000 data points. Then, instantaneous velocities were calculated using the averaged data. We further calculated the standard deviation of the instantaneous velocities for the part of the tether extension before elongation factors were injected into the chamber, and noted it as $\sigma_{pause}$. To distinguish between pausing and translocation, we use 2.5 $\sigma_{pause}$ as a threshold. All the absolute instantaneous velocities that are smaller than the threshold are attributed to pauses. All the absolute instantaneous velocities that are greater than or equal to the threshold are attributed to ribosome translocation (*Figure 1B–D*). The mean pause-free translocation velocity and translocation distance were calculated for each ribosome that actively translated the mRNA at constant forces. The mean translocation velocity for all the ribosomes that translate at specific constant forces was weighted by the total distance translocated by each ribosome. The mean translation velocities in nm/s were then converted to codon/s using the worm-like chain (WLC) model with a rise-per-base for ssRNA of 0.59 nm and a persistence length of 1 nm (*Liphardt et al., 2001*).

The translation rate vs force data was fitted with an exponential function using a weighted nonlinear least squares algorithm. Taking into account the possibility of a residual drift in the instrument, the stall-force was calculated as the force required to slow down the translation to a rate of 0.1 nt/s. Using the fitted exponential dependence parameters, this results in $F_{stall}$ = 13 ± 2 pN.

## Thermodynamic efficiency calculation

The free energy of ester hydrolysis was obtained as an average from the published values (*Fasman and Chemical Rubber Company, 1976*) of glycine ethyl ester (−8.40 kcal/mol), valyl RNA (−8.40 kcal/mol), and ethyl acetate (−4.72 kcal/mol), yielding −7.2 ± 1.2 kcal/mol. Similarly, the free energy of hydrolysis of amides was obtained as the average of published values (*Fasman and Chemical Rubber Company, 1976*) for asparagine (−3.60 kcal/mol), glutamine (−3.4 kcal/mol), to yield −3.5 ± 0.1 kcal/mol. Thus, the average energy available for mechanical work as a result of transpeptidation is ~3.7 ± 1.2 kcal/mol.

## Acknowledgements

AK, HFN and IT acknowledge support from the Human Frontier Science Program, and AK acknowledges support from the Israel Science Foundation, the European Commission, the Mallat Family Fund

and the Rubin Scientific and Medical Research Fund. This research was supported in part by the National Institutes of Health (NIH) grants GM-17129, GM-10840, and GM032543 (to HFN, IT and CB respectively), and by the US Department of Energy grant DE-AC0376SF00098(MSD KC261) to CB The content is solely the responsibility of the authors and does not necessarily represent the official views of the funding agencies.

## Additional information

### Funding

| Funder | Grant reference number | Author |
|---|---|---|
| Human Frontier Science Program | | Ariel Kaplan, Harry Noller, Ignacio Tinoco Jr |
| Israel Science Foundation | ISF 369/12 | Ariel Kaplan |
| European Commission Directorate-General for Research and Innovation | CIG 293923 | Ariel Kaplan |
| National Institutes of Health | GM032543 | Carlos J Bustamante |
| U.S. Department of Energy | DE-AC0376SF00098(MSD KC261) | Carlos J Bustamante |
| Mallat Family Fund | | Ariel Kaplan |
| National Institutes of Health | GM-10840 | Ignacio Tinoco Jr |
| National Institutes of Health | GM-17129 | Harry Noller |

The funders had no role in study design, data collection and interpretation, or the decision to submit the work for publication.

### Author contributions

TL, AK, Conception and design, Acquisition of data, Analysis and interpretation of data, Drafting or revising the article, Contributed unpublished essential data or reagents; LA, SY, CEW, Acquisition of data, Drafting or revising the article; J-DW, LL, Conception and design, Drafting or revising the article, Contributed unpublished essential data or reagents; KF, Conception and design, Contributed unpublished essential data or reagents; HN, IT, CJB, Conception and design, Analysis and interpretation of data, Drafting or revising the article

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
