## [Decision Letter]

Thank you for sending your work entitled “Direct Measurement of the Mechanical Work During Translocation by the Ribosome” for consideration at *eLife*. Your article has been favorably evaluated by Michael Marletta (Senior editor) and 2 reviewers, one of whom, Xiaowei Zhuang, is a member of our Board of Reviewing Editors.

The Reviewing editor has assembled the following comments to help you prepare a revised submission.

In this manuscript, the authors performed a single-molecule study of ribosome translocation using high-precision optical tweezers. They found that translocation rates depend exponentially on the applied apposing force. From the translocation rate vs. force curve, the authors determine the characteristic distance of the translocation or the distance between the transition state and the pre-translocation state and found the distance to be roughly one-codon length. From this result, the authors suggested that the translocation does not have sub-codon sub-steps. The authors further found that the translocation stalling force to be ∼13 pN, which is comparable to the force needed to unwind the most stable hairpin structures in mRNA, suggesting that the mRNA structure could be an efficient regulatory mechanism for translation. The translocation was also found to be interrupted by frequent pauses, with the pause-entry and pause-exit rates both being independent of the applied force.

These results represent a substantial advance of our understanding of how ribosomes function. The conclusions drawn in the work are supported by high quality data and thoughtful discussions. The Reviewing editor and the other reviewer both find the work exciting and recommend the publication of the manuscript in *eLife*. They only have a few minor points that they would like to see the authors address in the manuscript.

Minor comments:

1) Figure 1 shows that the translocation is interrupted by pauses and in between pauses, the translocation speed varies substantially. This variation was not only observed for different translocation bursts, but also within individual bursts. What is responsible for these variations?

2) Based on the observations in Figure 2, i.e. the translocation rate depending exponentially on the force, the authors concluded that the mRNA translocation itself is rate limiting. However, the result does not exclude the possibility that a conformational change effectively along the measurement coordinate is rate-limiting.

3) Based on the characteristic distance derived from the translocation rate vs. force curve, which is close to the length of a single codon, the authors concluded that the translocation occurs likely through a Brownian ratchet mechanism. It seems equally lightly that translocation occurs through a powerstroke mechanism with the transition state to be roughly one codon length sway from the pre-translocation state.

---

## [Author Response]

*1)*
Figure 1
*shows that the translocation is interrupted by pauses and in between pauses, the translocation speed varies substantially. This variation was not only observed for different translocation bursts, but also within individual bursts*. *What is responsible for these variations?*

Our data indicates that, in the conditions of our experiments, translocation is rate limiting to the overall translation reaction. As with any chemical reaction, which takes place when an intermediate, high-energy state is thermally reached, the time required for a single translocation step is a stochastic variable. Averaging over many such steps should result in a well-defined translocation rate. Most translation bursts observed in our experiments, however, are composed of a small number of such steps and, therefore, the stochastic nature of the basic step is still evident, resulting in bursts of different apparent rates. Note, that bursts containing a larger number of codons (such as the first and last in Figure. 1B, first panel) tend to be more uniform in their rates as a result of averaging.

*2) Based on the observations in*
Figure 2*, i.e. the translocation rate depending exponentially on the force, the authors concluded that the mRNA translocation itself is rate limiting. However, the result does not exclude the possibility that a conformational change effectively along the measurement coordinate is rate-limiting*.

The points of attachment of our molecular handles were chosen to minimize the ambiguity in this respect: First, by binding the 3' end of the mRNA from one side, and a small-subunit ribosomal protein on the other, we made sure that that our experiments are insensitive to any inter-subunit movements. Second, by choosing S16, in the “back” of the small subunit, which is far from any known functional site and not known to exhibit conformational dynamics during translocation, we ensure that the mechanical force affects and detects only movements between mRNA and the small subunit. By definition, such movement of the mRNA relative to the ribosome's small subunit, is the translocation step.

3) Based on the characteristic distance derived from the translocation rate vs. force curve, which is close to the length of a single codon, the authors concluded that the translocation occurs likely through a Brownian ratchet mechanism. It seems equally lightly that translocation occurs through a powerstroke mechanism with the transition state to be roughly one codon length sway from the pre-translocation state.

The reviewers are correct in mentioning that this possibility exists. A Brownian Ratchet (BR) mechanism implies that x˜=xstep, while a Power Stroke (PS) mechanism will require x˜<xstep. So, experimentally, when any measurement of x˜ contains some uncertainty because of the stochastic nature of the translocation time, the finite amount of data, and experimental limitations, it is impossible to completely rule out the PS mechanism. For that reason, we were careful in our manuscript not to state that the BR is the only possibility. We believe this is the most likely scenario because our best estimation of xbar (1.4) is extremely close to the step size (1.48). Notably, structural data (e.g. Zhou et. al., Science 2013) and kinetic experiments (e.g. Munro et. al., Nat. Struct. Mol. Biol. 2010) are consistent with this model.